# PRACTICAL $\epsilon$-EXPLORING THOMPSON SAMPLING FOR REINFORCEMENT LEARNING WITH CONTINUOUS CONTROLS

## ABSTRACT

Balancing exploration and exploitation is crucial in reinforcement learning (RL). While Thompson Sampling (TS) is a sound and effective exploration strategy, its application to RL with high-dimensional continuous controls remains challenging. We propose Practical $\epsilon$-Exploring Thompson Sampling (PETS), a practical approach that addresses these challenges. Since the posterior over the parameters of the action-value function is intractable, we leverage Langevin Monte Carlo (LMC) for sampling. We propose an approach which maintains $n$ parallel Markov chains to mitigate the issues of naïve application of LMC. The next step following the posterior sampling in TS involves finding the optimal action under the sampled model of the action-value function. We explore both gradient-based and gradient-free approaches to approximate the optimal action, with extensive experiments. Furthermore, to justify the use of gradient-based optimization to approximate the optimal action, we analyze the regret for TS in the RL setting with continuous controls and show that it achieves the best-known bound previously established for the discrete setting. Our empirical results demonstrate that PETS, as an exploration strategy, can be integrated with leading RL algorithms, enhancing their performance and stability on benchmark continuous control tasks.

## 1 INTRODUCTION

Reinforcement learning (RL) Mnih et al. (2015); Lillicrap et al. (2015); Sutton & Barto (1998) has become a cornerstone in solving complex decision-making problems, demonstrating significant success across diverse domains such as autonomous control (Kiumarsi et al., 2018), strategic game playing (Mnih et al., 2013b), and natural language processing Kung et al. (2022); Cetina et al. (2021). One of the central challenges of RL is striking a balance between exploration and exploitation (Chapelle & Li, 2011a; Auer, 2003; Berger-Tal et al., 2014; Nair et al., 2017). Exploration, the process of trying new actions to learn about their outcomes, is crucial for accurate value estimation and finding the optimal behaviour. Exploitation, on the other hand, involves leveraging existing knowledge to make optimal decisions. This exploration-exploitation dilemma is central to RL, profoundly influencing the learning process's efficiency and effectiveness (Sutton & Barto, 1998).

In continuous action spaces, the challenge of balancing exploration and exploitation is amplified due to the infinite number of possible actions. The volume of the action space grows exponentially with the number of dimensions, making efficient exploration in high-dimensional continuous spaces especially difficult (Tang et al., 2016). In such settings, simple and widely-used exploration methods like the $\epsilon$-greedy strategy (Tokic, 2010) fall short due to their inefficiency and lack of adaptability to knowledge acquired during learning (Dann et al., 2022). The maximum entropy framework introduces its own set of complications, in particular the difficulty of tuning the temperature hyperparameter $\alpha$ for entropy (Haarnoja et al., 2018b; Wang & Ni, 2020), which is crucial for balancing exploration and exploitation. The Upper Confidence Bound (UCB) method (Garivier & Moulines, 2011; Garivier & Cappé, 2011), while effective in the bandit setting, doesn't work well in practice in challenging continuous control tasks (Long & Han, 2023).

Thompson Sampling (TS) (Thompson, 1933) is an alternative exploration strategy that has been extensively explored in bandit problems (Chapelle & Li, 2011a; Xu et al., 2022). TS balances exploration and exploitation adaptively through probabilistic modeling of uncertainty. The essence of

TS lies in its principle of probability matching (Vulkan, 2000), i.e., the probability of selecting an action corresponds to the probability of that action being the optimal choice over the uncertainty of the knowledge about the environment. The more certain we are of our knowledge of the environment, the less TS explores. As learning progresses, TS naturally tends to favor actions that consistently yield better outcomes, thereby reducing exploration in areas where the understanding of the environment has solidified. In contrast, in contexts characterized by high uncertainty or sparse data, TS inherently boosts exploration to acquire more information. This adaptive approach enables TS to effectively manage the exploration-exploitation trade-off (Russo et al., 2017). When applied to the reinforcement learning (RL) setting, TS models the posterior over the parameters of the expected returns, the action-value function, for each state and action. It then selects the optimal action under a model of action-value function sampled from the posterior. In this approach, as new trajectories and their returns are observed, the uncertainty in the posterior decreases, and TS reduces exploration in favor of exploitation as a result (Saha & Kveton, 2023).

Despite Thompson Sampling's success in the bandit settings (Agrawal & Goyal, 2012; Slivkins et al., 2019; Kuleshov & Precup, 2014), its application to the more general RL setting has been limited. One of the main reasons is that the posterior, in all but the simplest cases, is intractable (van de Schoot et al., 2021). Consequently, sampling from the posterior, which is a necessary step in TS, is a challenge. To adapt TS to RL settings with high-dimensional continuous controls, we draw insights from Langevin Monte Carlo (LMC) (Langevin et al., 1908; Rossky et al., 1978; Roberts & Tweedie, 1996; Girolami & Calderhead, 2011). LMC provides a practical approach to sampling from intractable distributions in high-dimensional spaces.

However, naïvely applying LMC for posterior sampling does not fully resolve the challenges of performing TS in RL problems with continuous controls. (1) In TS, the step following posterior sampling involves finding the optimal action under the sampled model of the action-value function. In continuous action spaces, this task is non-trivial due to the infinite number of possible actions to consider. We explore both gradient-based and gradient-free optimization with extensive experiments for approximating the optimal action. Furthermore, to justify the use of gradient-based optimization, we analyze the regret for TS in the RL setting with continuous controls – to the best of our knowledge such analysis was previously limited to the discrete control setting (Ishfaq et al., 2024). We show that, under regularity conditions, the regret for TS with gradient-based optimization matches the best-known bound of $\widetilde{O}\left(d^{3/2}H^{3/2}\sqrt{T}\right)$ in the discrete setting. (2) Using samples from the LMC Markov chain at nearby steps can result in a high correlation between the sampled models of action-value function, which in turn leads to similar actions being explored. This is not ideal for effective exploration, as it limits the diversity of actions taken by the agent. Consequently, naïve application of LMC could lead to worse exploration compared to other exploration strategies because the posterior samples don't effectively represent the true posterior distribution. To tackle this issue, our approach, detailed in Section 3, involves maintaining $n$ parallel Markov chains. This helps us ensure a wider range of available posterior samples for action selection which results in better exploration. In Section 4.4, the effectiveness of this approach is empirically studied.

In this work, we introduce Practical $\epsilon$-Exploring Thompson Sampling (PETS), a practical algorithm that addresses the challenges that had previously limited the application of TS in challenging continuous control tasks. PETS can be incorporated into the existing RL approaches without requiring substantial modifications to their core algorithms. To demonstrate the effectiveness of our exploration strategy, we apply PETS to Policy Optimization with Model Planning (POMP) (Zhu et al., 2023), Model-Based Policy Optimization (MBPO) (Janner et al., 2019) and Soft Actor-Critic (SAC) (Haarnoja et al., 2018b) without modifying their inner workings and hyperparameters. We provide these results in Section 4 and show that our exploration strategy notably improves the results and stability of these methods.

## 1.1 CONTRIBUTIONS

Our contributions can be summarized as:

- We propose a TS-based exploration technique for RL with continuous controls.
- We explore gradient-based and gradient-free approaches to approximate the optimal action in TS. We conduct extensive experiments and provide further justification for the use of gradient-based optimization through a theoretical analysis of the regret.

- We introduce a practical approach for getting around the slow mixing in sampling from the posterior and reducing sample correlation, which we show results in better performance.

## 2 PRELIMINARIES

**Reinforcement Learning** We consider a discrete-time Markov Decision Process (MDP) (Puterman, 1994), represented by the tuple $(S, A, P, R, \gamma)$. Here, $S$ is the set of states, $A$ is the set of actions, $P(x'|x, a)$ is the transition probability, $R(x, a)$ is the one-step reward function, and $\gamma$ is the discount factor. The objective in RL is to find a policy $\pi$ that maximizes the expected cumulative discounted reward:

$$\max_{\pi} J(\pi) = \max_{\pi} \mathbb{E}_{\pi} \left[ \sum_{h=0}^{\infty} \gamma^h R(x_h, a_h) \right]$$

A policy $\pi : S \rightarrow P(A)$ maps states to a probability distribution over actions. The value function $V^{\pi}(x)$ is defined as the expected return starting from state $x$ under policy $\pi$:

$$V^{\pi}(x) = \mathbb{E}_{\pi} \left[ \sum_{h=0}^{\infty} \gamma^t R(x_h, a_h) | x_0 = x \right],$$

and the action-value function $Q^{\pi}(x, a)$ (Watkins & Dayan, 1992) represents the expected return for taking action $a$ in state $x$ and then following the policy $\pi$ afterwards:

$$Q^{\pi}(x, a) = \mathbb{E}_{\pi} \left[ \sum_{h=0}^{\infty} \gamma^t R(x_h, a_h) | x_0 = x, a_0 = a \right].$$

**Exploration vs. Exploitation** In reinforcement learning, exploration is fundamental for discovering optimal policies. It involves exploring the action space to gather more information about the environment, especially under uncertainty. Exploration can be viewed as a strategy where the probability selecting an action is not solely dependent on the current knowledge of the rewards (Watkins & Dayan, 1992), but also includes other components to encourage trying less-explored actions. This process is critical in environments with sparse or deceptive rewards, as it enables the agent to escape local optima and discover more rewarding strategies in the long run (Jiang et al., 2023). While exploration is key in learning about the environment, its counterpart, exploitation, is equally crucial (Wang et al., 2018) in RL. Exploitation involves leveraging the knowledge gained from exploration to make decisions that maximize immediate rewards. The balance between exploration and exploitation is a central challenge in RL, as excessive exploration can lead to sample inefficiency, while excessive exploitation might result in getting stuck at suboptimal policies. A well-calibrated balance ensures the agent learns effectively, adapts to the environment, and optimizes its strategy for long-term success.

**Thompson Sampling** Thompson Sampling (TS) (Russo et al., 2017) is a systematic approach to adaptively balance exploration and exploitation based on the uncertainty in the current knowledge about the environment. TS continuously updates a posterior distribution over the parameters of the model of expected returns, namely the action-value function $Q_w(x, a)$. In each step, TS first samples parameters $w$ from the posterior $p(w|\mathcal{D})$, where $\mathcal{D}$ is the observation set. Then, given the current state $x$, it selects an action $a^*$ that maximizes the expected return under the sampled model of the action-value function:

$$w \sim p(w|\mathcal{D}) \tag{1}$$

$$a^* = \arg\max_{a \in A} Q_w(x, a) \tag{2}$$

Crucially, as the posterior distribution reflects the uncertainty over the model of expected returns, TS inherently adjusts the exploration-exploitation trade-off by sampling from this distribution, with a higher uncertainty resulting in a greater exploration compared to exploitation.

## 3 METHOD

We model the cumulative return of taking an action $a$ at state $x$, $R_{x,a}$, with a Gaussian (Sutton & Barto, 1998), whose likelihood is denoted by $p(R_{x,a}|\mu_{x,a})$:

$$R_{x,a} \sim \mathcal{N}(\mu_{x,a}, 1) \tag{3}$$

where $\mu_{x,a}$ is the mean of the Gaussian. Typically in RL, the mean $\mu_{x,a}$ is modeled by using a function $Q_w$, parameterized by $w$, called the action-value function. Maximizing the likelihood of the observed returns under this Gaussian model is equivalent to minimizing the following objective:

$$L_{Q_w}(\mathcal{D}) = \mathbb{E}_{(x,a,r,x')\sim\mathcal{D}}\left[\left(Q_w(x,a) - R'_{x,a}\right)^2\right], \tag{4}$$

where $R'_{x,a}$ is the target return. The Q-learning objective (Watkins & Dayan, 1992) is recovered by setting the target return to

$$R'_{x,a} = r + \gamma \max_{a'} Q_w(x',a') \tag{5}$$

while Soft Actor-Critic (SAC) objective (Haarnoja et al., 2018a) is recovered by setting it to

$$R'_{x,a} = r + \gamma V_{w'}(x') \tag{6}$$

In machine learning, a commonly chosen prior for parameters is a Gaussian distribution with a zero mean and a variance of $\sigma_p^2$(Hoerl & Kennard, 2000). Under these choices of likelihood and prior:

$$-\log p(w|\mathcal{D}) = -\log p(\mathcal{D}|w) - \log p(w) + \log p(\mathcal{D}) \tag{7}$$

$$= \frac{1}{2}\sum_{i=1}^{N}\left[\left(Q_w(x_i,a_i) - R'_{x_i,a_i}\right)^2\right] + N\log(\sqrt{2\pi}) + \frac{1}{2\sigma_p^2}||w||^2 + \log(\sigma_p\sqrt{2\pi}) + \log p(\mathcal{D})$$

$$= \frac{1}{2}\mathbb{E}_{(x,a,r,x')\sim\mathcal{D}}\left[\left(Q_w(x_i,a_i) - R'_{x_i,a_i}\right)^2\right] + \frac{\lambda}{2}||w||^2 + C$$

where $C$ contains the constant terms and $\lambda = \frac{1}{\sigma_p^2}$. Consider the following objective:

$$\mathcal{L}_{Q_w}(\mathcal{D}) = \mathbb{E}_{(x,a,r,x')\sim\mathcal{D}}\left[\left(Q_w(x,a) - R'_{x,a}\right)^2\right] + \lambda||w||^2 \tag{8}$$

where the choice of $\lambda$ determines how informative the prior is, with $\lambda = 0$ corresponding to the least informative prior, i.e., uniform distribution.

By Eq 7 we have $\mathcal{L}_{Q_w}(\mathcal{D}) \propto -\log p(w|\mathcal{D})$ and consequently:

$$p(w|\mathcal{D}) = \frac{1}{Z}\exp(-\mathcal{L}_{Q_w}(\mathcal{D})) \tag{9}$$

where $Z$ is the partition function, also known as the normalizing constant, necessary to ensure that $p(w|\mathcal{D})$ integrates to 1.

### 3.1 Leveraging Langevin Monte Carlo

To dynamically balance exploration and exploitation, we use Thompson Sampling. This requires sampling from the posterior distribution described in Eq 9. However, the partition function $Z$, except for trivial cases, is intractable. One way to sample from the posterior without needing to compute $Z$ is Markov chain Monte Carlo (MCMC)(Levin & Peres, 2017; Holden, 2019; Sahlin, 2011) sampling. A common Markov chain used in MCMC sampling is Langevin dynamics (Langevin et al., 1908; Lemons & Gythiel, 1997) which is characterized by a stochastic differential equation (SDE) defined as:

$$dw(s) = -\nabla L(w(s))ds + \sqrt{2\beta^{-1}}dB(s), \tag{10}$$

where $L$ is an objective function parameterized by $w$, $s$ is a continuous time index, $B$ is a Brownian motion, and $\beta$ is an inverse temperature parameter. The Euler-Maruyama (Faniran, 2015) discretization of this equation, also known as Langevin Monte Carlo (LMC) (Rossky et al., 1978; Girolami & Calderhead, 2011; Durmus et al., 2018) is given by:

$$w_{t+1} = w_t - \eta_t \nabla L_t(w_t) + \sqrt{2\beta_t^{-1}\eta_t}\epsilon_t, \tag{11}$$

where $\eta_t$ is the learning rate at time step $t$, and $\epsilon_t$ is isotropic Gaussian noise. This discretization enables LMC to approximate the continuous-time process of Eq 10. One can use a mini batch of observed data instead of a full batch to compute the gradients, giving rise to the famous stochastic gradient Langevin dynamics (SGLD) (Welling & Teh, 2011). Under certain conditions, Eq 11

generates a Markov chain whose marginal distribution converges to a unique distribution $p(w) \propto \exp(-\beta L(w))$ (Zou et al., 2020; Roberts & Tweedie, 1996).

Stochastic gradient Langevin dynamics (SGLD) offers a practical solution to sampling from the posterior without explicitly computing the partition function. To sample, one needs to apply LMC Eq 11 for an adequate number of iterations in order for the Markov chain to mix (Levin & Peres, 2017).

## 3.2 PARALLEL POSTERIOR SAMPLES

While SGLD provides a sampling method without needing to compute the partition function $Z$, it results in highly correlated samples (Vishnoi, 2021) at nearby steps of Eq 11. This results in similar actions in the TS procedure, which is suboptimal in environments requiring a high degree of exploration and diversity in decision-making.

To avoid that, one needs to discard intermediate samples, also known as the burn-in period (Sahlin, 2011). In other words, the Markov chain generated by Eq 11 should be run for many steps (long burn-in period) to adequately mix (Levin & Peres, 2017; Holden, 2019; Sahlin, 2011). However, in challenging continuous control tasks where the agent needs to take actions over the course of hundreds of thousands of steps, running this Markov chain to mix for every action sample is highly inefficient.

To address this challenge, instead of maintaining a single Markov chain, which requires many steps to mix, or using nearby samples that result in high correlation, we maintain $n$ independent Markov chains, $\mathcal{W} = \{w^{(1)}, w^{(2)}, \ldots, w^{(n)}\}$ where each $w^{(i)}$ is trained on different batches from the replay buffer (see line 21 to 24 of Algorithm 1). For action decisions, we randomly select one of these $n$ posterior samples, $w_{\text{selected}} \sim \text{Uniform}(\mathcal{W})$ where $\text{Uniform}(\mathcal{W})$ indicates a discrete uniform distribution over the elements in the set $\mathcal{W}$.

This approach ensures a representative exploration of the posterior distribution resulting in a more diverse set of actions and better exploration. Moreover, another practical advantage of this approach is that the Markov chains in $\mathcal{W}$ can be trained independently in parallel, improving computational efficiency. In Section 4.4 we empirically validate the effectiveness of this approach in achieving better exploration.

## 3.3 APPROXIMATING THE OPTIMAL ACTION IN CONTINUOUS SPACES

Leveraging LMC to sample from the posterior does not fully address the challenges of using TS in RL with continuous controls. A remaining challenge is that the step following the posterior sampling involves finding the optimal action w.r.t the model of the action-value function, as described by Eq 2. This is straightforward in discrete action spaces but becomes challenging in high-dimensional continuous action spaces, as there are infinitely many possible actions to consider. We explore both gradient-based and gradient-free optimization approaches. We use Adam for gradient-based optimization, and one of the more recent methods, design by adaptive sampling (DBAS) (Brookes & Listgarten, 2018), for gradient-free optimization. Experimental results in Section 4.2 demonstrate that PETS with both gradient-based and gradient-free optimization outperforms the baselines, including the state-of-the-art RL algorithm in the continuous control setting, POMP (Zhu et al., 2023).

### 3.3.1 DBAS

Given the sampled model of the action-value function $Q_{w_{\text{selected}}}$ and the current state, our objective is to find the optimal action $a^*$ as described by Eq 2. Design by adaptive sampling (DBAS) (Brookes & Listgarten, 2018) is a gradient-free, iterative algorithm that can be used to approximate the optimal action.

In each iteration $i$ of DBAS: (I) It trains an unconditional generative model $G_i$ on a set of actions $\mathcal{A}_i$, where $\mathcal{A}_0$ can be initialized randomly or from a policy. (II) It samples a new set of actions from $G_i$ and initialize $\mathcal{A}_{i+1}$ with them. (III) It uses $Q_{w_{\text{selected}}}$ as an oracle to rank the samples in $\mathcal{A}_{i+1}$, retaining the top $k$ actions in $\mathcal{A}_{i+1}$—where $k$ is a hyperparameter—and discarding the rest. This process is repeated for $n$ iterations, where $n$ is a hyperparameter. Finally, one of the actions in $\mathcal{A}_n$ can be used as an approximation to the optimal action. In Appendix B.1, we provide the implementation details and pseudocode for this procedure. We refer readers to Brookes & Listgarten (2018) for a more detailed description.

### 3.3.2 GRADIENT-BASED OPTIMIZATION

Gradient-based optimization is an iterative approximation process:

$$a_{t+1} = a_t + \eta \nabla_a Q_{w_{\text{selected}}}(x, a_t), \tag{12}$$

where $a_t$ is the action at iteration $t$, $\eta$ is the learning rate, and $\nabla_a Q_{w_{\text{selected}}}(x, a_t)$ is the gradient of the action-value function with respect to the action, $Q_{w_{\text{selected}}}$ is the action-value function parameterized by the selected posterior sample $w_{\text{selected}}$, and $x$ is the current state. The initial action $a_0$ can either be initialized randomly or sampled from the policy.

This approach provides a practical approximation for finding the optimal action in continuous action spaces. To complement our experimental results and justify the use of gradient-based optimization, we analyze the regret under the setting with linear MDP(Puterman, 1994) and linear function approximation, as is standard in the literature (Ishfaq et al., 2024; Zhang et al., 2021; Wang et al., 2020). Under this setting, replacing the exact optimal action with an approximate optimal action found by gradient-based optimization yields Algorithm 3. Our analysis culminates in Theorem 3.1, which shows that under regularity conditions, the regret for Algorithm 3 matches the best-know regret bound for TS in the discrete control setting (Ishfaq et al., 2024). Below we state the main result of our analysis informally. Rigorous definitions and proofs are available in Appendix C.

**Theorem 3.1.** *Under appropriate choices of $\lambda$ in Eq 7, $\beta$ in Eq 11, learning rate and update number for LMC in Eq 11, if the action-value function is L-smooth and satisfies the Polyak-Łojasiewicz (PL) inequality (Karimi et al., 2020), the regret of Algorithm 3 satisfies*

$$\text{Regret}(K) = \widetilde{O}\left(d^{3/2} H^{3/2} \sqrt{T}\right), \tag{13}$$

*with probability at least $1 - \delta$ where $\delta \in (\frac{1}{2\sqrt{2e\pi}}, 1)$.*

Furthermore, in Theorem C.11 we show that with the additional cost of extending the parameter space, $w$, it is possible to achieve the same bound with a high probability of $1 - \epsilon'$ for any $\epsilon' \in (0, 1)$.

### 3.4 IMPLEMENTATION DETAILS

We draw insights from Jin et al. (2023) and incorporate an $\epsilon$ parameter into our TS algorithm. Particularly, with probability $\epsilon$, we use TS to select an action, and with probability $1 - \epsilon$ we use the underlying RL algorithm to select an action (see Algorithm 1). Jin et al. (2023) demonstrates that $\epsilon$-TS improves the computational efficiency of TS while achieving better regret bounds across several reward functions. For optimizations, we use the Adam optimizer (Kingma, 2014) in all cases. PETS-specific hyperparameters are provided in Appendix B.

## 4 EXPERIMENTS

In our experiments, we aim to study four primary questions: (1) Can PETS, as a general exploration strategy, be integrated into recent RL algorithms and enhance their performance? (2) How does the integrated PETS perform compared to state-of-the-art RL algorithms in challenging continuous control tasks? (3) Does PETS lead to better exploration? and (4) How effective is the approach of maintaining multiple parallel posterior samples for achieving better exploration and results? In order to answer these questions, in the following sections, we conduct experiments on a range of continuous control tasks from OpenAI Gym benchmark suite (Brockman et al., 2016). For a fair comparison we keep all components and hyperparameters of the underlying RL algorithms, POMP, MBPO, and SAC, the same in all cases.

### 4.1 GENERALITY OF PETS

To address the first question, we integrate PETS into the leading RL algorithm in the continuous setting, POMP (Zhu et al., 2023), resulting in PETS-POMP. Additionally, we integrate PETS into two other recent algorithms: A high-performing model-based algorithm, MBPO (Janner et al., 2019), and a well-established model-free algorithm, SAC (Haarnoja et al., 2018b) resulting in PETS-MBPO and PETS-SAC respectively. In all cases, we maintain the methodologies and hyperparameters of the underlying algorithms, POMP, MBPO, and SAC. The comparison between PETS-POMP and POMP can be viewed in Figure 3. As shown in the figure, PETS-POMP outperforms POMP on several challenging continuous control tasks. Further, in Figure 1 and 2 we compare the performance

---

**Algorithm 1** PETS Pseudocode

---

**Input:** exploration probability $\epsilon$, number of posterior samples $n_{\text{samples}}$, learning rates $\eta$ and $\eta'$, whether to use gradient-based or gradient-free optimization $use\_grad$, gradient-free procedure $\mathcal{Y}$, number of gradient ascent steps $n_{\text{grad\_steps}}$, environment $\mathcal{E}$, RL algorithm $\mathcal{A}$

1: $\mathcal{W} = \{w_1, w_2, \ldots, w_{n_{\text{samples}}}\}$ and $w_i \sim \mathcal{N}(\vec{0}, I)$ {Initialize posterior samples randomly}
2: Initialize the replay buffer $\mathcal{B} \leftarrow \emptyset$
3: **repeat**
4:     Observe current state $x$ from environment $\mathcal{E}$
5:     Draw a random value $p$ from Uniform(0, 1)
6:     **if** $p < \epsilon$ **then**
7:         $w_{\text{selected}} \sim \text{Uniform}(\mathcal{W})$
8:         **if** $use\_grad$ **then**
9:             Initialize action $a$ randomly or by following $\mathcal{A}$'s procedure
10:             **for** $j = 1$ to $n_{\text{grad\_steps}}$ **do**
11:                 $a \leftarrow a + \eta' \nabla_a Q_{w_{\text{selected}}}(x, a)$
12:             **end for**
13:         **else**
14:             $a \leftarrow \mathcal{Y}(Q_{w_{\text{selected}}})$ {(see Algorithm 2)}
15:         **end if**
16:     **else**
17:         Find action $a$ by following $\mathcal{A}$'s procedure
18:     **end if**
19:     Take action $a$ in environment $\mathcal{E}$, observe next state $x'$ and reward $r$
20:     $\mathcal{B} \leftarrow \mathcal{B} \cup \{(x, a, r, x')\}$ {Add observation to the replay buffer}
21:     **for** each sample $w_i$ in $\mathcal{W}$ **do**
22:         Draw a batch of observations from $B \sim \mathcal{B}$
23:         $w_i \leftarrow w_i - \eta \nabla_{w_i} \mathcal{L}_{Q_{w_i}}(B) + \sqrt{2\beta^{-1}\eta}\epsilon_t$ {SGLD where $\epsilon_t$ is an isotropic Gaussian noise}
24:     **end for**
25:     Update RL algorithm $\mathcal{A}$ with observations from $\mathcal{B}$
26: **until** convergence criterion is met

---

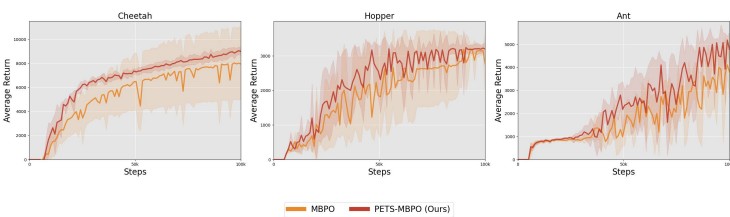

Figure 1: Learning curves of PETS-MBPO (Ours) and MBPO on three continuous control tasks. The solid lines represent the mean and the shaded areas represent the standard deviation among trials of 5 different seeds. The MBPO hyperparameters for PETS-MBPO and MBPO are the same across these experiments. As shown in this figure, our exploration strategy improves the results of MBPO across several tasks.

of PETS-MBPO with MBPO and PETS-SAC with SAC on several continuous control tasks. We observe that in both cases, our exploration strategy improves the performance of MBPO and SAC on several tasks. These sets of results demonstrate the effectiveness and generalizability of PETS in improving the results and stability of different existing RL algorithms and its potential as an effective exploration strategy.

### 4.2 COMPARISON WITH BASELINES

To address the second question, we evaluate the performance of PETS-POMP compared to six leading model-free and model-based RL algorithms. As shown in Figure 3, both PETS-POMP with gradient-free and PETS-POMP with gradient-based optimization outperform all the baselines on several challenging tasks. Notably, our method achieves better performance on Humanoid and Ant, which generally are considered to be the most challenging OpenAI Gym (Todorov et al., 2012; Brockman et al., 2016) tasks. Specifically, our method improves POMP's results by 38%, 29%, and 11% on

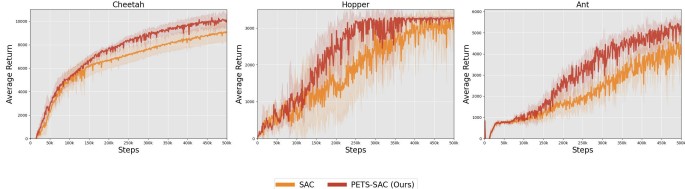

Figure 2: Learning curves of PETS-SAC (Ours) and SAC on three continuous control tasks. The solid lines represent the mean and the shaded areas represent the standard deviation among trials of 5 different seeds. The SAC hyperparameters for PETS-SAC and SAC are the same across these experiments. As shown in this figure, our exploration strategy improves the results of SAC across several tasks.

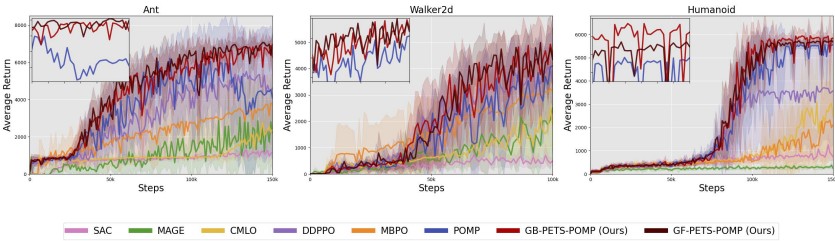

Figure 3: Learning curves of PETS-POMP (Ours) and baselines on three of the most challenging continuous control tasks from OpenAI Gym benchmark. GB-PETS-POMP and GF-PETS-POMP correspond to our method using gradient-based and gradient-free optimization approaches, respectively, for approximating the optimal action. The solid lines represent the mean and the shaded areas represent the standard deviation among trials of 8 different seeds. Both implementations of our method achieve better performance and training stability compared to the baselines. Specifically, our method improves POMP's results by $38\%$, $29\%$, and $11\%$ on Walker2d, Ant, and Humanoid, respectively. In each sub-figure, the small upper-left plot shows the zoomed-in comparison of our method and POMP during the final iterations.

Walker2d, Ant, and Humanoid, respectively. This demonstrates the effectiveness of our method in challenging continuous control tasks. Figure 6 further illustrates PETS-POMP's performance on three additional tasks: InvertedPendulum, Hopper, and Cheetah. The implementation details and hyperparameter settings for all experiments are described in Appendix B.

### 4.3 PETS EXPLORATION EFFECTIVENESS

In this section, to further investigate the reasons for PETS's superior performance address the third question, we visualize the diversity of actions taken by our exploration policy compared to POMP's policy (Zhu et al., 2023). To achieve this, we take PETS actions for $40$ steps in the Hopper environment, resulting in a trajectory of length $40$. Starting from the same initial state, we take POMP actions for the same number of steps. This process is repeated 30 times, resulting in 30 trajectories for each method. In Figure 4, we visualize the standard deviation of the hopper's height ($z$ coordinate) at each step across these trajectories. PET's actions result in a wider range of height changes, demonstrating a greater diversity in the outcomes of actions taken by PETS compared to POMP. This greater diversity, a result of PETS' exploration, correlates with its superior performance demonstrated in Figures 3 and 6.

### 4.4 ABLATION

In this section, we try to address the fourth question by investigating the effectiveness of maintaining parallel posterior samples by conducting two ablation studies. First, we vary the number of parallel posterior samples, $n$, and observe the returns in Figure 5a. As shown in the figure, larger values of $n$ result in a higher return. Second, we compare the results of maintaining $n$ parallel posterior samples to the case where we maintain only one posterior sample with a burn-in period (Sahlin, 2011) of $n$ where $n-1$ intermediate samples in Eq 11 are discarded as discussed in Section 3. Figure 5b shows that maintaining multiple parallel samples results in a higher return compared to using a single posterior sample with a longer burn-in period. As higher returns reflect a more effective exploration, these ablation studies show that maintaining multiple parallel posterior samples leads to better exploration by ensuring a more representative exploration of the posterior distribution.

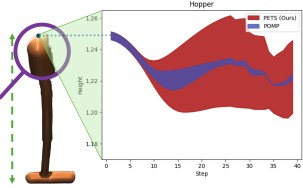

Figure 4: Visualization of action diversity for PETS and POMP in the Hopper environment. On the left, the hopper's state in the first iteration is shown. On the right, the range of changes in the hopper's height ($z$ coordinate) is visualized over 40 steps, for PETS actions in red and POMP actions in blue. To achieve this, we take PETS actions in the Hopper environment for 40 steps, generating a trajectory of length 40. Starting from the same initial state, we take POMP actions for the same number of steps. This process is repeated 30 times for both PETS and POMP, resulting in 30 trajectories for each method. At each step, the standard deviation of the hopper's height across these trajectories is visualized. As shown, PETS demonstrates greater diversity in the outcomes of its actions, correlating with its superior performance compared to POMP.

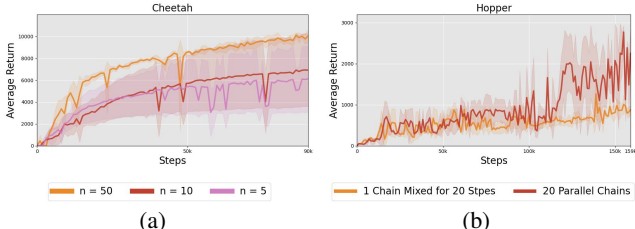

Figure 5: **(a)** Learning curves of PETS-MBPO for three different values for number of parallel posterior samples, $n$, averaged over three trials with different seeds in the Cheetah environment. As shown, a larger value of $n$ results in higher returns. This demonstrates the effectiveness of our approach to improve exploration by maintaining a wide range of posterior samples. **(b)** Comparison of maintaining multiple parallel posterior samples with the case of maintaining only one with a burn-in period of $n$ for the Markov chain generated by Eq 11. The curves are averaged over three trials with different seeds in the Hopper environment. As shown, maintaining multiple parallel posterior samples leads to higher returns compared to maintaining one, demonstrating the capability of this approach to generate less correlated posterior samples.

## 5 RELATED WORK

**Exploration.** As one of the most central research problems in RL, exploration has been extensively studied. In this section, we discuss some of the works in this field, while recognizing that the breadth of literature on exploration is far too vast to be comprehensively covered here. In the context of discrete action spaces: Osband et al. (2014) shows that least-square value iteration using $\epsilon$-greedy is highly inefficient. By introducing randomized least-square value iteration (RLSVI), they show that randomized value functions can be efficient and effective. Fortunato et al. (2017) adds noise to value and action-value functions with learned coefficients to boost exploration. Auer (2003) encourages exploration in the bandit setting by using confidence bounds and introducing bonuses for less-visited arms. Henaff et al. (2023) proposes to make use of both global and episodic bonuses to improve exploration. Chen et al. (2017) proposes an exploration strategy based on UCB by leveraging uncertainty estimates from the Q-ensemble to boost exploration. Jarrett et al. (2022) proposes to learn representations of the future that capture the unpredictable aspects of each outcome and use that as additional input for predictions. Pislar et al. (2021) draws inspiration from animals and humans and proposes mode-switching exploration in RL where they introduce an approach to adaptively switch between modes of exploration. Furthermore, among the exploration approaches in the context of continuous action spaces: Bellemare et al. (2016) uses density models to measure uncertainty and proposes a method to derive a pseudo-count from an arbitrary density model. Lobel et al. (2023) proposes a count-based exploration strategy suited for high-dimensional state spaces. To this end, they estimate visitation counts by averaging samples from the Rademacher distribution. Pathak et al. (2019) trains an ensemble of dynamics models and incentivizes the agent to explore such that the disagreement of those ensembles is maximized.

**Thompson Sampling.** TS has been extensively studied in the multi-armed bandit (Slivkins et al., 2019; Kuleshov & Precup, 2014) setting. (Agrawal & Goyal, 2012) showed that TS achieves a logarithmic expected regret and it is competitive to or better than UCB (Garivier & Moulines, 2011; Garivier & Cappé, 2011). Komiyama et al. (2015) adapted TS to the setting where multiple arms need to be selected at the same time. In the contextual bandit setting, Agrawal et al. (2017) applies TS in scenarios with linear payoffs and proves a high probability regret bound of $\tilde{O}\left(\frac{d^2}{\epsilon}\sqrt{T^{1+\epsilon}}\right)$. Xu et al. (2022) demonstrates that Laplace approximation (Chapelle & Li, 2011b) of the posterior distribution is inefficient in high-dimensional settings. They leverage Langevin Monte Carlo (Langevin et al., 1908) for posterior sampling and show that TS achieves a regret bound of $\tilde{O}(d\sqrt{dT})$. Moreover, Jin et al. (2023) introduces $\epsilon$-Exploring Thompson Sampling ($\epsilon$-TS) which selects arms based on the posterior with a probability of $\epsilon$. They show that $\epsilon$-TS improves the computational efficiency of TS while achieving better regret bounds. They further demonstrate the superiority of $\epsilon$-TS for a range of reward distributions, such as Bernoulli, Gaussian, Poisson, and Gamma. Ishfaq et al. (2024) demonstrates the effectiveness of Langevin Monte Carlo Thompson Sampling (LMC-TS) in Atari games (Mnih et al., 2013a) where the discrete action space contains at most 18 possible actions. They prove that LMC-TS achieves the regret bound of $\tilde{O}(d^{3/2}H^{3/2}\sqrt{T})$ in the linear MDP setting with discrete actions under specific assumptions. While they demonstrate the effectiveness of TS in Atari games (Mnih et al., 2013a), they do not address the challenges limiting the application of TS in challenging continuous control tasks.

**High-dimensional Continuous Control.** While the mentioned challenges in Section 3.2 have limited the application of TS in high-dimensional continuous control tasks, a great amount of progress has been made by other methods, not necessarily focusing on the exploration/exploitation trade-off. Among model-free algorithms, Schulman et al. (2015) improves policy updates by ensuring small, incremental changes, using a trust region to maintain policy performance and stability by using a surrogate objective function. Schulman et al. (2017) introduces a simple surrogate objective function that lower-bounds the performance of a policy. In contrast to TRPO (Schulman et al., 2015), PPO (Schulman et al., 2017) only requires first-order information to optimize the policy. SAC (Haarnoja et al., 2018a) tries to address the exploration/exploitation trade-off using the Maximum Entropy Reinforcement Learning framework. On the other hand, model-based approaches have shown great progress in improving the sample efficiency and performance of RL algorithms. POMP (Zhu et al., 2023) incorporates Deep Differential Dynamic Programming (D3P) planner into the model-based RL and shows significant improvement on MuJoCo tasks. MAGE (D'Oro & Jaśkowski, 2020) leverages the environment model differentiability to directly compute policy gradients. MBPO (Janner et al., 2019) makes use of the environment model with different horizons and shows that their approach matches the asymptotic performance of the best model-free algorithms. CMLO (Ji et al., 2022) proposes an event-triggered mechanism to determine when to update the model of the environment. DDPPO (Li et al., 2022) proposes a two-model-based learning method to control the prediction and gradient error.

## 6 CONCLUSION

In this work, we introduced Practical $\epsilon$-Exploring Thompson Sampling (PETS), which aims to address the challenges that have limited the application of Thompson Sampling (TS) in RL with continuous control tasks. We draw insights from Langevin Monte Carlo (LMC) for posterior sampling and propose an approach to maintain $n$ parallel posterior samples, mitigating the issue of sample correlation. Additionally, we employ both gradient-based and gradeient-free approaches to approximate the optimal action and provide theoretical guarantees that under regularity conditions, replacing the exact optimal action with an approximate one found by gradient-based optimization achieves the best-known regret bound of $\widetilde{O}\left(d^{3/2}H^{3/2}\sqrt{T}\right)$. To validate PETS's effectiveness as an exploration strategy, we integrated it into POMP (Zhu et al., 2023), MBPO (Janner et al., 2019) and SAC (Haarnoja et al., 2018b). Our empirical results demonstrate PETS's effectiveness in improving the performance and training stability of existing RL alorithms across a range of challenging continuous control tasks.

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

# A  ADDITIONAL RESULTS

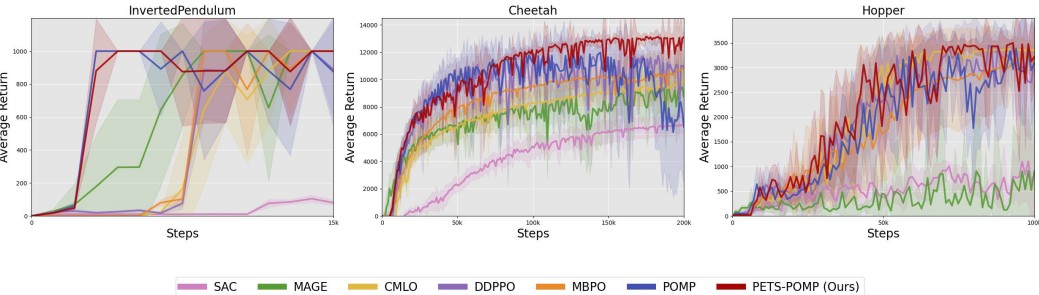

Figure 6: Learning curves of PETS-POMP (Ours) and baselines on three continuous control tasks from OpenAI Gym benchmark. The solid lines represent the mean and the shaded areas represent the standard deviation among trials of 8 different seeds. As shown in the figure, our method achieves better performance compared to baselines on InvertedPendulum and Cheetah while achieving comparable results on Hopper. Moreover, PETS-POMP outperforms POMP on all tasks while also improving its training stability.

# B  IMPLEMENTATION DETAILS

In this section, we provide some details for the implementation of our method.

## B.1  DBAS DETAILS

Here we provide more implementation details for DBAS (Brookes & Listgarten, 2018). The pseudocode for DBAS procedure can be found in Algorithm 2. In our method, we use a mixture of Gaussians as the generative model $G$. We use the Expectation-Maximization (E-M) algorithm (Moon, 1996) to train the mixture of Gaussians. Furthermore, we use an initial set of actions drawn from the policy for $x_{\text{init}}$. We repeat the DBAS procedure (Line 5 of Algorithm 2) for 10-20 iterations.

---

**Algorithm 2** DBAS

**Input:** predictor oracle $O(x)$, GenTrain$(x_i)$, percentage of least-performing samples $[q = 0.9]$, number of samples $[M = 1000]$, initial data set $[x_{\text{init}} = \emptyset]$

1: $set \leftarrow x_{\text{init}}$
2: **if** $x_{\text{init}}$ is empty **then**
3:     $set \leftarrow$ randomly initialized data
4: **end if**
5: **while** not converged **do**
6:     $G \leftarrow$ GenTrain$(set)$
7:     $set \leftarrow x_i \sim G$
8:     $scores_i \leftarrow O(x_i)$
9:     $set \leftarrow x_i$ if it is not among the $q^{th}$ percentage least-performing samples based on $scores$
10: **end while**
11: return $set_0$

---

## B.2  HYPERPARAMETERS

In Table 1 and Table 2 and 3 we present the set of hyperparameters used in PETS-POMP, PETS-MBPO and PETS-SAC respectively.

## B.3 ACTION INITIALIZATION

When using gradient-based optimization to find the approximate optimal action, action initialization can make a difference in the quality of the approximation. This also can be seen in our regret analysis in Appendix C. We empirically have found that initializing the action with an action sampled from the policy helps with getting better approximations.

Table 1: Set of hyperparameters used in PETS-POMP.

|  |  | Inverted Pendulum | Walker2d | Cheetah | Ant | Humanoid | Hopper |
|---|---|---|---|---|---|---|---|
| $\epsilon$ | exploration probability | 0.6 | 0.3 | 0.8 | 0.3 | 0.8 | 0.8 |
| $n_{\text{samples}}$ | number of posterior samples | 5 | 5 | 50 | 10 | 5 | 10 |
| $n_{\text{grad\_steps}}$ | number of gradient ascent steps | 40 | 100 | 100 | 60 | 100 | 50 |
| $\eta'$ | gradient ascent learning rate | 0.002 | 0.01 | 0.01 | 0.05 | 0.05 | 0.005 |

Table 2: Set of hyperparameters used in PETS-POMP.

|  |  | Cheetah | Ant | Hopper |
|---|---|---|---|---|
| $\epsilon$ | exploration probability | 0.8 | 0.4 | 0.4 |
| $n_{\text{samples}}$ | number of posterior samples | 50 | 5 | 5 |
| $n_{\text{grad\_steps}}$ | number of gradient ascent steps | 100 | 40 | 40 |
| $\eta'$ | gradient ascent learning rate | 0.01 | 0.01 | 0.01 |

Table 3: Set of hyperparameters used in PETS-SAC.

|  |  | Cheetah | Ant | Hopper |
|---|---|---|---|---|
| $\epsilon$ | exploration probability | 0.7 | 0.4 | 1.0 |
| $n_{\text{samples}}$ | number of posterior samples | 5 | 5 | 5 |
| $n_{\text{grad\_steps}}$ | number of gradient ascent steps | 20 | 40 | 80 |
| $\eta'$ | gradient ascent learning rate | 0.02 | 0.01 | 0.01 |

## C  REGRET ANALYSIS

We expand upon the regret analysis presented in Ishfaq et al. (2024), extending it to approximate greedy policies. We demonstrate that under certain assumptions on the action-value function, the regret bound of $\widetilde{O}\left(d^{3/2}H^{3/2}\sqrt{T}\right)$ can be achieved, even when the optimal action $a^*$ cannot be trivially identified and has to be approximated.

To this end, we first state the analysis setting. Consider an episodic MDP of the form $(S, \mathcal{A}, H, \mathbb{P}, r)$ where $S$ is the state space, $\mathcal{A}$ is the continuous action space, $H$ is the episode length, $\mathbb{P} = \{\mathbb{P}_h\}_{h=1}^{H}$ are the state transition probability distributions, and $r = \{r_h\}_{h=1}^{H}$ are the reward functions where $r_h : S \times A \to [0, 1]$.

Furthermore, $\pi_h(x)$ denotes the action that the agent takes in the state $x$ at the $h$-th step in the episode, and $\pi$ is the set of policies. The value and action-value functions are defined as:

$$V_h^\pi(x) = \mathbb{E}_\pi\left[\sum_{h'=h}^H r_{h'}(x_{h'}, a_{h'}) \mid x_h = x\right].$$

$$Q_h^\pi(x, a) = \mathbb{E}_\pi\left[\sum_{h'=h}^H r_{h'}(x_{h'}, a_{h'}) \mid x_h = x, a_h = a\right].$$

The Bellman equation and Bellman optimality equations are as follows:

$$Q_h^\pi(x, a) = \left(r_h + \mathbb{P}_h V_{h+1}^\pi\right)(x, a), \quad V_h^\pi(x) = Q_h^\pi(x, \pi_h(x)), \quad V_{H+1}^\pi(x) = 0.$$

$$Q_h^*(x, a) = \left(r_h + \mathbb{P}_h V_{h+1}^*\right)(x, a), \quad V_h^*(x) = Q_h^*(x, \pi_h^*(x)), \quad V_{H+1}^*(x) = 0.$$

where $V_h^*(x) = V_h^{\pi^*}(x)$, $Q_h^*(x, a) = Q_h^{\pi^*}(x, a)$, $\pi^*$ is the optimal policy and $[\mathbb{P}_h V_{h+1}](x, a) = \mathbb{E}_{x' \sim \mathbb{P}_h(\cdot|x,a)} V_{h+1}(x')$.

We measure the suboptimality of an agent by the total regret defined as

$$\text{Regret}(K) = \sum_{k=1}^K \left[V_1^*\left(x_1^k\right) - V_1^{\pi^k}\left(x_1^k\right)\right] \tag{14}$$

where $x_1^k$ is the initial state and $\pi_k$ is the policy agent uses for episode $k$. and $K$ is the total number of episodes which the agent interacts with the environment with the goal of learning the optimal policy.

Consider the following loss function for the action-value function from Ishfaq et al. (2024):

$$L_h^k(w_h) = \sum_{\tau=1}^{k-1}\left[r_h(x_h^\tau, a_h^\tau) + \max_{a \in \mathcal{A}} Q_{h+1}^k(x_{h+1}^\tau, a) - Q(w_h; \phi(x_h^\tau, a_h^\tau))\right]^2 + \lambda\|w_h\|^2 \tag{15}$$

where $\phi(.,.)$ is a feature vector, $w_h$ is the Q function parameters and $\lambda$ is the regularization constant. We consider a linear function approximation for the Q function and:

$$Q_h^k(\cdot, \cdot) \leftarrow \min\left\{\phi(\cdot, \cdot)^\top w_h^{k, J_k}, H - h + 1\right\}^+ \tag{16}$$

where $w_h^{k, J_k}$ is the parameter vector obtained after $J_k$ iterations of the Langevin Monte Carlo (LMC) process, applied to the loss function defined in Eq. equation 17, as described in Algorithm 3. We further denote $V_h^k(x_h^k) = \max_{a \in \mathcal{A}} Q_h^k(x_h^k, a)$.

Note that while the action-value function $Q_h^k$ is linear with respect to the parameter vector $w$, it is not necessarily linear in the action $a$. Furthermore, the loss function $L_h^k(w_h)$ includes the term $V_{h+1}^k(x_{h+1}^\tau)$, which, in a high-dimensional continuous action space, cannot be computed exactly due to infinite actions. Consequently, in Algorithm 3 and our regret analysis, we substitute this term with the approximate value function, $\hat{V}_h^k$, as defined in Equation equation 12. This approach leads to the formulation of the following modified loss function:

$$L_h^k(w_h) = \sum_{\tau=1}^{k-1}\left[r_h(x_h^\tau, a_h^\tau) + \hat{V}_{h+1}^k(x_{h+1}^\tau) - Q(w_h; \phi(x_h^\tau, a_h^\tau))\right]^2 + \lambda\|w_h\|^2 \tag{17}$$

We present a modified version of the Langevin Monte Carlo Least Squares Value Iteration (LMC_LSVI) algorithm (Ishfaq et al., 2024) in Algorithm 3. Contrasting with Algorithm 1 in Ishfaq et al. (2024), Algorithm 3 incorporates the use of an approximate value function, denoted as $\hat{V}$, and employs approximate optimal actions. In the subsequent sections, we demonstrate that under certain assumptions about the action-value function, Algorithm 3 achieves the same target regret bound.

---

**Algorithm 3** Langevin Monte Carlo Least-Squares Value Iteration (LMC-LSVI) with Approximate Greedy Policy

---

**Input:** step sizes $\{\eta_k > 0\}_{k \geq 1}$, inverse temperature $\{\beta_k\}_{k \geq 1}$, loss function $L_k(w)$

1: Initialize $w_h^{1,0} = \mathbf{0}$ for $h \in [H]$, $J_0 = 0$
2: **for** episode $k = 1, 2, \ldots, K$ **do**
3:     Receive the initial state $s_1^k$
4:     **for** step $h = H, H-1, \ldots, 1$ **do**
5:        $w_h^{k,0} = w_h^{k-1,J_{k-1}}$
6:        **for** $j = 1, \ldots, J_k$ **do**
7:           $\epsilon_h^{k,j} \sim \mathcal{N}(0, I)$
8:           $w_h^{k,j} = w_h^{k,j-1} - \eta_k \nabla L_h^k \left( w_h^{k,j-1} \right) + \sqrt{2\eta_k \beta_k^{-1}} \epsilon_h^{k,j}$
9:        **end for**
10:       $Q_h^k(\cdot, \cdot) \leftarrow \min \left\{ Q \left( w_h^{k,J_k}; \phi(\cdot, \cdot) \right), H - h + 1 \right\}^+$
11:       initialize set of actions $a_{h,0}^k$
12:       **for** iteration $t = 1, 2, \ldots, t^*$ **do**
13:          $a_{h,t}^k = a_{h,t-1}^k + \nabla Q(., a_{h,t-1}^k)$ {as outlined in Eq 12}
14:       **end for**
15:       $\hat{V}_h^k(\cdot) \leftarrow Q(., a_{h,t^*}^k)$ {approximate optimal value}
16:     **end for**
17:     **for** step $h = 1, 2, \ldots, H$ **do**
18:       Take approximate optimal action $a_h^k$ based on $a_{h,t^*}^k$, observe reward $r_h^k(s_h^k, a_h^k)$ and next state $s_{h+1}^k$
19:     **end for**
20: **end for**

---

**Proposition C.1.** *As defined in Ishfaq et al. (2024), let $w^{k,J_k}$ be the approximation of posterior parameters after $J_k$ iterations of LMC as defined in Eq 11 for the k'th episode where h is the horizon step. Under the loss defined in Eq 17, $w^{k,J_k}$ follows a Gaussian distribution where the mean vector and covariance matrix are defined as:*

$$\mu_h^{k,J_k} = A_k^{J_k} \ldots A_1^{J_1} w_h^{1,0} + \sum_{i=1}^{k} A_k^{J_k} \ldots A_{i+1}^{J_{i+1}} \left( I - A_i^{J_i} \right) \widehat{w}_h^i, \qquad (18)$$

$$\Sigma_h^{k,J_k} = \sum_{i=1}^{k} \frac{1}{\beta_i} A_k^{J_k} \ldots A_{i+1}^{J_{i+1}} \left( I - A_i^{2J_i} \right) \left( \Lambda_h^i \right)^{-1} \left( I + A_i \right)^{-1} A_{i+1}^{J_{i+1}} \ldots A_k^{J_k}, \qquad (19)$$

*where $A_i = I - 2\eta_i \Lambda_h^i$ for $i \in [k]$.*

*Proof.* We refer readers to Proposition B.1 of Ishfaq et al. (2024) for the proof. $\qquad \square$

**Definition C.2.** (Model Prediction Error). For any $(k, h) \in [K] \times [H]$, we define the model prediction error associated with the reward $r_h$,

$$l_h^k(x, a) = r_h(x, a) + \mathbb{P}_h \hat{V}_{h+1}^k(x, a) - Q_h^k(x, a). \qquad (20)$$

**Proposition C.3.** *For an action-value function $Q(x, .)$ that has an L-Lipschitz continuous gradient and satisfies the Polyak-Łojasiewicz Inequality (PL) inequality for some $\mu > 0$ as stated below:*

$$\frac{1}{2} \|\nabla Q_h^k(x, a)\|^2 \geq \mu \left( Q_h^k(x, a) - Q_h^k(x, a^*) \right), \quad \forall a. \qquad (21)$$

*where*

$$a^* = argmax_a Q_h^k(x_h, a)$$

*is the optimal action. For $t^* >= \left( \frac{L}{\mu} log(KH \frac{Q_h^k(x_h, a^*) - Q_h^k(x_h, a_0)}{d^{3/2} H^{3/2} \sqrt{T}}) \right)$ iterations we have:*

$$V_h^k(x_h) - \hat{V}_h^k(x_h) \leq \varepsilon_{t^*} \qquad (22)$$

*where $\varepsilon_{t^*} = \frac{d^{3/2} H^{3/2} \sqrt{T}}{KH}$.*

*Proof.* As proved in Theorem 1. of Karimi et al. (2020), with the step size of $\frac{1}{L}$, for an action-value function $Q(x, .)$ that has an $L$-Lipschitz continuous gradient and satisfies the PL inequality, we have

$$Q_h^k(x_h, a^*) - Q_h^k(x_h, a_t) \leq \left(1 - \frac{\mu}{L}\right)^t \left(Q_h^k(x_h, a^*) - Q_h^k(x_h, a_0)\right) \tag{23}$$

where $a_t$ is the action after $t$ iteration of gradient ascent:

$$a_{t+1} = a_t + \nabla Q_h^k(x_h, a_t) \tag{24}$$

using $1 - u \leq \exp(-u)$ on Eq 23 we have

$$Q_h^k(x_h, a^*) - Q_h^k(x_h, a_t) \leq \exp\left(-t\frac{\mu}{L}\right)\left(Q_h^k(x_h, a^*) - Q_h^k(x_h, a_0)\right).$$

So for $t^* >= \frac{L}{\mu} log(KH \frac{Q_h^k(x_h, a^*) - Q_h^k(x_h, a_0)}{d^{3/2} H^{3/2} \sqrt{T}})$ we have:

$$Q_h^k(x_h, a^*) - Q_h^k(x_h, a_{t^*}) \leq \frac{d^{3/2} H^{3/2} \sqrt{T}}{KH} = \varepsilon_{t^*} \tag{25}$$

by applying the definition of $V_h^k$ we have:

$$V_h^k(x_h) - \hat{V}_h^k(x_h) \leq \varepsilon_{t^*} \tag{26}$$

$\square$

**Proposition C.4.** *Under the approximate value function $\hat{V}_h^k$ we have:*

$$\mathbb{P}_h V_{h+1}^k(x, a) - \mathbb{P}_h \hat{V}_{h+1}^k(x, a) \leq \varepsilon_{t^*} \tag{27}$$

*where $\varepsilon_{t^*}$ is defined in Proposition C.3*

*Proof.* Applying the definition of $\mathbb{P}_h$ we have:

$$[\mathbb{P}_h V_{h+1}](x, a) = \mathbb{E}_{x' \sim \mathbb{P}_h(\cdot|x,a)} V_{h+1}(x') \leq \mathbb{E}_{x' \sim \mathbb{P}_h(\cdot|x,a)} \hat{V}_{h+1}(x') + \varepsilon_{t^*} \tag{28}$$

$$= \left[\mathbb{P}_h \hat{V}_{h+1}\right](x, a) + \varepsilon_{t^*} \tag{29}$$

where the first to the second line is by using Proposition C.3. $\square$

**Lemma C.5.** *Let $\lambda = 1$ in Eq 17, Define the following event*

$$\mathcal{E}(K, H, \delta) = \left\{ \left| \phi(x, a)^\top \hat{w}_h^k - r_h(x, a) - \mathbb{P}_h V_{h+1}^k(x, a) \right| \right.$$
$$\left. \leq 5H\sqrt{d}C_\delta \|\phi(x, a)\|_{(\Lambda_h^k)^{-1}}, \forall (h, k) \in [H] \times [K] \text{ and } \forall (x, a) \in \mathcal{S} \times \mathcal{A} \right\}. \tag{30}$$

*where we denote*

$$C_\delta = \left[\frac{1}{2} \log(K + 1) + \log\left(\frac{2\sqrt{2}KB_{\delta/2}}{H}\right) + \log\frac{2}{\delta}\right]^{1/2}$$

*and $B_\delta = \left(\frac{16}{3}Hd\sqrt{K} + \sqrt{\frac{2K}{3\beta_K \delta}}d^{3/2}\right)$. Then we have $\mathbb{P}(\mathcal{E}(K, H, \delta)) \geq 1 - \delta$.*

*Proof.* We refer readers to Lemma. B5. of (Ishfaq et al., 2024) for the proof. $\square$

**Lemma C.6.** *Let $\lambda = 1$ in Eq 17. For any $\delta \in (0, 1)$ conditioned on the event $\mathcal{E}(K, H, \delta)$, for all $(h, k) \in [H] \times [K]$ and $(x, a) \in \mathcal{S} \times \mathcal{A}$, with probability at least $1 - \delta^2$, we have*

$$-\left(r_h(x, a) + \mathbb{P}_h V_{h+1}^k(x, a) - Q_h^k(x, a)l_h^k(x, a)\right) \tag{31}$$

$$\leq \left(5H\sqrt{d}C_\delta + 5\sqrt{\frac{2d\log(1/\delta)}{3\beta_K}} + 4/3\right)\|\phi(x, a)\|_{(\Lambda_h^k)^{-1}}, \tag{32}$$

*where $C_\delta$ is defined in Lemma C.5.*

*Proof.* We refer readers to Lemma. B.6 of (Ishfaq et al., 2024) for the proof. □

**Lemma C.7.** *(Error bound). Let $\lambda = 1$ in Eq 17. For any $\delta \in (0,1)$ conditioned on the event $\mathcal{E}(K, H, \delta)$, for all $(h, k) \in [H] \times [K]$ and $(x, a) \in \mathcal{S} \times \mathcal{A}$, with probability at least $1 - \delta^2$, we have*

$$-l_h^k(x, a) \leq \left(5H\sqrt{d}C_\delta + 5\sqrt{\frac{2d\log(1/\delta)}{3\beta_K}} + 4/3\right) \|\phi(x, a)\|_{(\Lambda_h^k)^{-1}} + \varepsilon_{t^*}, \tag{33}$$

*Proof.* using Lemma C.6 and Proposition C.4 we have:

$$- \left(r_h(x, a) + \mathbb{P}_h \hat{V}_{h+1}^k(x, a) - Q_h^k(x, a).l_h^k(x, a) + \varepsilon t^*\right)$$

$$= l_h^k(x, a) - \varepsilon t^* \leq \left(5H\sqrt{d}C_\delta + 5\sqrt{\frac{2d\log(1/\delta)}{3\beta_K}} + 4/3\right) \|\phi(x, a)\|_{(\Lambda_h^k)^{-1}}$$

□

**Lemma C.8.** *Let $\lambda = 1$ in Eq 17. Conditioned on the event $\mathcal{E}(K, H, \delta)$, for all $(h, k) \in [H] \times [K]$ and $(x, a) \in \mathcal{S} \times \mathcal{A}$, with probability at least $\frac{1}{2\sqrt{2e\pi}}$, we have*

$$- \left(r_h(x, a) + \mathbb{P}_h V_{h+1}^k(x, a) - Q_h^k(x, a)\right) \leq 0 \tag{34}$$

*Proof.* We refer readers to Lemma. B.7 of (Ishfaq et al., 2024) for the proof. □

**Lemma C.9.** *(Optimism). Let $\lambda = 1$ in Eq 17. Conditioned on the event $\mathcal{E}(K, H, \delta)$, for all $(h, k) \in [H] \times [K]$ and $(x, a) \in \mathcal{S} \times \mathcal{A}$, with probability at least $\frac{1}{2\sqrt{2e\pi}}$, we have*

$$l_h^k(x, a) \leq 0 \tag{35}$$

*Proof.* We immediately get the stated result by using Proposition C.4 on Lemma C.8. □

We restate the main theorem:

**Theorem C.10.** *Let $\lambda = 1$ in Eq 17, $\frac{1}{\beta_k} = \widetilde{O}(H\sqrt{d})$ in Algorithm 3 and $\delta \in \left(\frac{1}{2\sqrt{2e\pi}}, 1\right)$. For any episode $k \in [K]$, let the learning rate $\eta_k = 1/\left(4\lambda_{\max}\left(\Lambda_h^k\right)\right)$, the update number for LMC in Eq 11 be $J_k = 2\kappa_k \log(4HKd)$ where $\kappa_k = \lambda_{\max}\left(\Lambda_h^k\right)/\lambda_{\min}\left(\Lambda_h^k\right)$ is the condition number of $\Lambda_h^k$ defined in Proposition C.1. Under the assumption that the action-value function $Q_h^k$ in Eq 16 has an L-Lipschitz continuous gradient and satisfies the Polyak-Łojasiewicz Inequality (PL) inequality Eq 21, the regret of Algorithm 3 under the regret definition in Definition 14, satisfies*

$$\text{Regret}(K) = \widetilde{O}\left(d^{3/2}H^{3/2}\sqrt{T}\right), \tag{36}$$

*with probability at least $1 - \delta$.*

Proof of Theorem C.10. By Lemma. 4.2 in (Cai et al., 2020), it holds that

$$\text{Regret}(T) = \sum_{k=1}^{K} \left(V_1^*\left(x_1^k\right) - V_1^{\pi^k}\left(x_1^k\right)\right) \tag{37}$$

$$= \underbrace{\sum_{k=1}^{K}\sum_{t=1}^{H} \mathbb{E}_{\pi^*}\left[\langle Q_h^k\left(x_h, \cdot\right), \pi_h^*\left(\cdot \mid x_h\right) - \pi_h^k\left(\cdot \mid x_h\right)\rangle \mid x_1 = x_1^k\right]}_{(i)} + \underbrace{\sum_{k=1}^{K}\sum_{t=1}^{H} \mathcal{D}_h^k}_{(ii)} \tag{38}$$

$$+ \underbrace{\sum_{k=1}^{K}\sum_{t=1}^{H} \mathcal{M}_h^k}_{(iii)} + \underbrace{\sum_{k=1}^{K}\sum_{h=1}^{H} \left(\mathbb{E}_{\pi^*}\left[l_h^k\left(x_h, a_h\right) \mid x_1 = x_1^k\right] - l_h^k\left(x_h^k, a_h^k\right)\right)}_{(iv)}, \tag{39}$$

where $\langle .,. \rangle$ denotes inner product which in continuous spaces is defined as $\langle f, g \rangle = \int_D f(t)g(t)\,dt$. Furthermore, $\mathcal{D}_h^k$ and $\mathcal{M}_h^k$ are defined as

$$\mathcal{D}_h^k := \left\langle \left(Q_h^k - Q_h^{\pi^k}\right)\left(x_h^k, \cdot\right), \pi_h^k\left(\cdot, x_h^k\right) \right\rangle - \left(Q_h^k - Q_h^{\pi^k}\right)\left(x_h^k, a_h^k\right),$$

$$\mathcal{M}_h^k := \mathbb{P}_h\left(\left(V_{h+1}^k - V_{h+1}^{\pi^k}\right)\right)\left(x_h^k, a_h^k\right) - \left(V_{h+1}^k - V_{h+1}^{\pi^k}\right)\left(x_h^k\right).$$

**Bounding Term (i):** Using Proposition C.3 we have $Q(x_h, a^*) - Q(x_h, a_{t^*}) \le \varepsilon_{t^*}$.

Note that $\pi_h^k$ is approximately greedy w.r.t $Q_h^k$ and $\pi_h^k(a_{t^*}) = 1$ where $a_{t^*}$ is the approximate optimal greedy action from Eq 25. The largest value that $\left\langle Q_h^k\left(x_h, \cdot\right), \pi_h^*\left(\cdot \mid x_h\right) - \pi_h^k\left(\cdot \mid x_h\right)\right\rangle$ in Eq 38 can take is $Q(x_h, a^*) - Q(x_h, a_{t^*})$ which happens if $\pi_h^*(a^*) = 1$ where $a^* = \operatorname{argmax}_a Q_h^k(x_h, a)$. This completes the proof using Eq 25.

**Bound for Term (ii):** With probability $1 - \delta/3$ we have:

$$\sum_{k=1}^{K}\sum_{h=1}^{H}\mathcal{D}_h^k \le \sqrt{2H^2 T \log(3/\delta)} \tag{40}$$

We refer the readers to the Appendix. B.2 of (Ishfaq et al., 2024) for the proof.

**Bound for Term (iii):** With probability $1 - \delta/3$ we have:

$$\sum_{k=1}^{K}\sum_{h=1}^{H}\mathcal{M}_h^k \le \sqrt{2H^2 T \log(3/\delta)}. \tag{41}$$

We refer the readers to the Appendix. B.2 of Ishfaq et al. (2024) for the proof.

**Bound for Term (iv):** With probability at least $\left(1 - \frac{\delta}{3} - \frac{1}{2\sqrt{2e\pi}}\right)$ we have:

$$\sum_{k=1}^{K}\sum_{h=1}^{H}\left(\mathbb{E}_{\pi^*}\left[l_h^k\left(x_h, a_h\right) \mid x_1 = x_1^k\right] - l_h^k\left(x_h^k, a_h^k\right)\right) \le \widetilde{O}\left(d^{3/2} H^{3/2}\sqrt{T}\right) \tag{42}$$

Suppose the event $\mathcal{E}\left(K, H, \delta'\right)$ holds. by union bound, with probability $1 - \left(\delta'^2 + \frac{1}{2\sqrt{2e\pi}}\right)$, we have,

$$\sum_{k=1}^{K}\sum_{h=1}^{H}\left(\mathbb{E}_{\pi^*}\left[l_h^k\left(x_h, a_h\right) \mid x_1 = x_1^k\right] - l_h^k\left(x_h^k, a_h^k\right)\right) \tag{43}$$

$$\leq \sum_{k=1}^{K}\sum_{h=1}^{H} -l_h^k\left(x_h^k, a_h^k\right) \tag{44}$$

$$\leq \sum_{k=1}^{K}\sum_{h=1}^{H}\left(5H\sqrt{d}C_{\delta'} + 5\sqrt{\frac{2d\log\left(1/\delta'\right)}{3\beta_K}} + 4/3\right)\left\|\phi\left(x_h^k, a_h^k\right)\right\|_{\left(\Lambda_h^k\right)^{-1}} + KH\varepsilon_{t^*} \tag{45}$$

$$= \left(5H\sqrt{d}C_{\delta'} + 5\sqrt{\frac{2d\log\left(1/\delta'\right)}{3\beta_K}} + 4/3\right)\sum_{k=1}^{K}\sum_{h=1}^{H}\left\|\phi\left(x_h^k, a_h^k\right)\right\|_{\left(\Lambda_h^k\right)^{-1}} + KH\varepsilon_{t^*} \tag{46}$$

$$\leq \left(5H\sqrt{d}C_{\delta'} + 5\sqrt{\frac{2d\log\left(1/\delta'\right)}{3\beta_K}} + 4/3\right)\sum_{h=1}^{H}\sqrt{K}\left(\sum_{k=1}^{K}\left\|\phi\left(x_h^k, a_h^k\right)\right\|_{\left(\Lambda_h^k\right)^{-1}}^2\right)^{1/2} + KH\varepsilon_{t^*} \tag{47}$$

$$\leq \left(5H\sqrt{d}C_{\delta'} + 5\sqrt{\frac{2d\log\left(1/\delta'\right)}{3\beta_K}} + 4/3\right)H\sqrt{2dK\log(1+K)} + KH\varepsilon_{t^*} \tag{48}$$

$$= \left(5H\sqrt{d}C_{\delta'} + 5\sqrt{\frac{2d\log\left(1/\delta'\right)}{3\beta_K}} + 4/3\right)\sqrt{2dHT\log(1+K)} + KH\varepsilon_{t^*} \tag{49}$$

$$= \left(5H\sqrt{d}C_{\delta'} + 5\sqrt{\frac{2d\log\left(1/\delta'\right)}{3\beta_K}} + 4/3\right)\sqrt{2dHT\log(1+K)} + \left(d^{3/2}H^{3/2}\sqrt{T}\right) \tag{50}$$

$$= \widetilde{O}\left(d^{3/2}H^{3/2}\sqrt{T}\right). \tag{51}$$

Here the first, the second, and the third inequalities follow from Lemma C.9, Lemma C.7 and the Cauchy-Schwarz inequality respectively. The last inequality follows from Lemma C.5 The last equality follows from $\frac{1}{\sqrt{\beta_K}} = 10H\sqrt{d}C_{\delta'} + \frac{8}{3}$ which we defined in Lemma C.9. By Lemma C.7, the event $\mathcal{E}\left(K, H, \delta'\right)$ occurs with probability $1 - \delta'$. Thus, by union bound, the event $\mathcal{E}\left(K, H, \delta'\right)$ occurs and it holds that

$$\sum_{k=1}^{K}\sum_{h=1}^{H}\left(\mathbb{E}_{\pi^*}\left[l_h^k\left(x_h, a_h\right) \mid x_1 = x_1^k\right] - l_h^k\left(x_h^k, a_h^k\right)\right) \leq \widetilde{O}\left(d^{3/2}H^{3/2}\sqrt{T}\right)$$

By applying union bound for (i), (ii), (iii) and (iv), the final regret bound is $\widetilde{O}\left(d^{3/2}H^{3/2}\sqrt{T}\right)$ with at least probability $1 - \delta$ where $\delta \in \left(\frac{1}{2\sqrt{2e\pi}}, 1\right)$.

**Theorem C.11.** *Let $\lambda = 1$ in Eq 17, $\frac{1}{\beta_k} = \widetilde{O}(H\sqrt{d})$ in Algorithm 3 and $\delta \in \left(\frac{1}{2\sqrt{2e\pi}}, 1\right)$. For any episode $k \in [K]$, let the learning rate $\eta_k = 1/\left(4\lambda_{\max}\left(\Lambda_h^k\right)\right)$, the update number for LMC in Eq 11 be $J_k = 2\kappa_k\log(4HKd)$ where $\kappa_k = \lambda_{\max}\left(\Lambda_h^k\right)/\lambda_{\min}\left(\Lambda_h^k\right)$ is the condition number of $\Lambda_h^k$ defined in Proposition C.1. Let $\vec{w} = [w_1, w_2, \ldots, w_n]^T$ be the extended parameter space and $Q(x, a) = \max_{i\in[n]} Q_{w_i}(x, a)$ be the optimistic action-value function. Under the assumption that the action-value function $Q_h^k$ in Eq 16 has an L-Lipschitz continuous gradient and satisfies the Polyak-Łojasiewicz Inequality (PL) inequality Eq 21, the regret of Algorithm 3 under the regret definition in Definition 14, satisfies*

$$\text{Regret}(K) = \widetilde{O}\left(d^{3/2}H^{3/2}\sqrt{T}\right), \tag{52}$$

*with probability of $1 - \epsilon'$ for any $\epsilon' \in (0, 1)$.*

Proof of Theorem C.11. In Lemma B.9 we prove that the estimation $Q_h^k(x, a)$ is optimistic with a constant probability of at least $\frac{1}{2\sqrt{2e\pi}}$. In other words, the failure probability is at most $1 - \frac{1}{2\sqrt{2e\pi}}$. By extending the parameter space $\vec{w} = [w_1, w_2, \ldots, w_n]^T$ and modelling the optimistic action-value function using $Q(x, a) = \max_{i \in [n]} Q_{w_i}(x, a)$, the failure probability will be at most $(1 - \frac{1}{2\sqrt{2e\pi}})^n$. We want this probability to be arbitrarily small. To guarantee that the failure probability is less than $\epsilon'$ it suffices to find an $n$ that is large enough such that $(1 - \frac{1}{2\sqrt{2e\pi}})^n < \epsilon'$. If we solve for $n$ we have $n > \frac{\log \epsilon'}{\log(1 - \frac{1}{2\sqrt{2e\pi}})}$. We can express the latter quantity as $\frac{\log(1/\delta)}{\log(2\sqrt{2e\pi}) - \log(2\sqrt{2e\pi} - 1)} \in \Omega(\log(1/\epsilon'))$. So, we can extend the parameter space by a factor of $\Omega(\log(1/\epsilon'))$ to ensure that the failure probability is less than $\epsilon'$. Finally, we can apply the union bound on (i), (ii), (iii), and (iv) to conclude that the regret bound in Theorem C.11 holds with a probability of $1 - \epsilon'$ for any $\epsilon' \in (0, 1)$.

