# OpenReview forum: "Practical $\epsilon$-Exploring Thompson Sampling for Reinforcement Learning with Continuous Controls"
_ICLR.cc/2025/Conference — ICLR 2025 Conference Withdrawn Submission_

### Official Review · Reviewer_Lyym · 2024-10-20

**Soundness:** 3
**Presentation:** 3
**Contribution:** 3
**Rating:** 6
**Confidence:** 2

**Summary:**

PETS introduces a new exploration strategy for RL in continuous control tasks, leveraging TS while addressing its challenges in high-dimensional environments. In traditional bandit problems, TS has proven effective, but scaling it to RL with continuous actions is complex because the posterior distribution is intractable in most cases. To address the intractability of posterior sampling in TS, PETS uses Langevin Monte Carlo, which is a method for generating samples from complex distributions. However, naive application of LMC often results in highly correlated samples, which limits the diversity of explored actions. PETS mitigates this by running parallel Markov chains to obtain a wider range of posterior samples, improving exploration. Once the posterior sample is obtained, the next challenge in continuous action spaces is to select the optimal action under the sampled model of the action-value function. PETS explores both gradient-based and gradient-free approaches to optimize action selection. The theoretical analysis in PETS shows that gradient-based optimization achieves regret bounds comparable to the best-known results for discrete settings, making it a viable choice for continuous control. PETS was tested with popular RL algorithms like Policy Optimization with Model Planning (POMP), Model-Based Policy Optimization (MBPO), and Soft Actor-Critic (SAC), demonstrating improved exploration and stability without requiring major modifications to the underlying algorithms.

**Strengths:**

PETS enhances exploration in RL by maintaining multiple posterior samples and using them to explore a wider range of potential actions. This parallel ensemble approach helps in achieving better exploration of the state-action space. PETS has been shown to outperform state-of-the-art RL algorithms in complex continuous control tasks, such as Walker2d, Ant, and Humanoid, achieving higher returns and more stable learning curves. Its exploration mechanism is particularly beneficial for environments with high-dimensional and continuous action spaces. PETS leverages Thompson Sampling principles to minimize regret. Theoretical analysis shows that it matches the best-known regret bounds in certain settings, such as in linear Markov Decision Processes (MDPs) with continuous actions, ensuring its long-term learning efficiency.

**Weaknesses:**

PETS maintains multiple posterior samples and performs parallel exploration, which can increase the computational burden, especially when scaling to high-dimensional environments. The need to maintain and update multiple models or samples can lead to higher memory and CPU/GPU requirements, particularly when dealing with large state and action spaces. While PETS uses Thompson Sampling to balance exploration and exploitation, it might still struggle in environments with sparse rewards or deceptive reward structures. The method may over-explore in certain cases, leading to inefficient learning when exploitation would yield better results. Similarly, the $\epsilon$-Exploring Thompson Sampling ($\epsilon$-TS) might lead to suboptimal performance when $\epsilon$ is not well-calibrated.

**Questions:**

PETS has demonstrated success in linear Markov Decision Processes (MDPs). Do you have an idea of its performance in highly non-linear or non-stationary environments ?
The algorithm’s theoretical guarantees for regret minimization are often more straightforward in linear settings, what is the bottleneck in non-linear settings ?
Are there specific problem types or domains where PETS is particularly effective?

---

### Official Review · Reviewer_P91A · 2024-10-29

**Soundness:** 2
**Presentation:** 2
**Contribution:** 1
**Rating:** 3
**Confidence:** 5

**Summary:**

The paper discusses how to apply Langevin Monte Carlo (LMC) based Thompson sampling algorithm for continuous action space RL tasks. Relying on the result of Ishfaq et al 2024, the authors derive a regret bound for continuous action setting and also provide some experiments. The paper further explores both gradient based and gradient free methods for finding approximate optimal action.

**Strengths:**

1. The work is nice in the sense that it extends LMC-LSVI from Ishfaq et al 2024a to continuous action setting which is an important problem setting. It also shows the efficacy of the method through some thorough experiment.

2. The proposal of parallel posterior sampling is a promising way to mitigate the issue of correlated sampling in vanilla LMC.

**Weaknesses:**

1. My main complaint for this work is that it doesn’t situate the contribution of the paper relative to other highly related papers such as Ishfaq et al 2024a and Ishfaq et al 2024b. In line 70-74, it introduces the idea of using LMC for TS as if it’s the first work to do so. Proposition C.1, Definition C.2, Lemma C.5,  Lemma C.6, Lemma C.7, Lemma C.8, Lemma C.9 all are essentially verbatim to different lemmas used in the proof of Ishfaq et al 2024a. With this respect the novelty of the algorithmic contribution and theoretical analysis are very limited in this paper.

2. Practical $\epsilon$ Exploring Thompson Sampling (PETS) directly borrows ideas/methods from two existing works $\epsilon$-TS [Jin et al 2023] and LMC-LSVI of Ishfaq et al 2024a. As such the novelty in terms of algorithmic design seems limited. Despite borrowing significant chunk of Ishfaq 2024a for algorithm design, the paper fails to properly acknowledge it in their method section, especially in Section 3.1.

3. Also, I think all the assumptions made in the theorem statements such as L-smoothness and Polyak Lojasiewicz inequality in Theorem 3.1 should be clearly defined in the main paper.

4. Missing related work: I think in Section 5, around Line 486, the authors should discuss other provably efficient randomized exploration methods such as Ishfaq et al 2021, Russo 2019, Xiong et al 2022 etc.

5. Missing code: Given the empirical aspect of the work, I think the authors should share the code even during the reviewing process. Could you please share the code base anonymously?

6. Mujoco experiments: for mujoco experiments in Fig3, it was run for 150k or 100k time steps whereas the standard is to run for 1million step. Could you explain why this is the case? With this low number of steps, the result interpretation can be unfair. Also, can you consider stronger baseline such as DSAC-T (Duan et al 2023)?


Ishfaq, Haque, et al. "Provable and Practical: Efficient Exploration in Reinforcement Learning via Langevin Monte Carlo." The Twelfth International Conference on Learning Representations. 2024a

Ishfaq, Haque, et al. "More Efficient Randomized Exploration for Reinforcement Learning via Approximate Sampling" The 1st Reinforcement Learning Conference. 2024b

Ishfaq, Haque, et al. "Randomized exploration in reinforcement learning with general value function approximation." International Conference on Machine Learning. PMLR, 2021.

Daniel Russo. Worst-case regret bounds for exploration via randomized value functions. 2019

Xiong et al. Near-optimal randomized exploration for tabular Markov decision processes 2022

Duan et al. DSAC-T: Distributional Soft Actor-Critic with Three Refinements, 2023

**Questions:**

1. In line 303, it says “For optimizations, we use the Adam optimizer in all cases.” But in Line 23 of Algorithm 1, the update rule is standard SGLD update. Could you please clarify how exactly Adam is used here? Also, do you use Adam LMCDQN based adaptive bias term as explained in Ishfaq et al 2024a?

2. Could you comment on how the idea of maintaining n parallel Markov chains is different from the optimistic sampling scheme mentioned in Remark 4.3 and Appendix D.1 in Ishfaq et al. 2024a?

---

### Official Review · Reviewer_cPiZ · 2024-10-31

**Soundness:** 3
**Presentation:** 3
**Contribution:** 2
**Rating:** 5
**Confidence:** 3

**Summary:**

The authors propose a TS-based exploration technique for RL with continuous action spaces. To manage the continuous action space, they use Langevin Monte Carlo (LMC) to sample from the posterior of the action-value function and approximate the optimal action through both gradient-based and gradient-free methods. They provide a regret guarantee for linear MDPs and demonstrate empirical performance improvements in continuous control tasks by integrating their PETS approach with established RL algorithms.

**Strengths:**

Integrating RL theory with deep RL is a fascinating and crucial direction for advancing the RL community. The application of theoretically proven techniques to real-world problems, with demonstrated success, is especially compelling. This made the paper both exciting and highly engaging to read. I believe this type of work will make a significant contribution to the community.

**Weaknesses:**

- My main concern is the technical novelty. The proof of Theorem 1 seems very similar to the analysis in Ishfaq et al., 2024. What is the main technical challenge or novelty in this work compared to Ishfaq et al., 2024, particularly in proving Theorem 1? A detailed discussion of these technical challenges in the main paper would be very helpful.

- Regarding the experiments, I have several comments:
    1. Generally, a strong exploration strategy is particularly beneficial in sparse reward settings. However, it appears that all the experiments in this paper were conducted in dense reward environments. Do you believe your algorithm would still demonstrate improved performance in sparse reward settings?

    2. The paper lacks an analysis of computational costs. The proposed method appears to be computationally intensive due to the large number of posterior samples and gradient ascent steps, especially in the Cheetah environment where $n=50$, likely resulting in considerable computational overhead. Including a comparison of computational costs with baseline algorithms would be helpful for understanding the contribution of your work.


    3. I wonder if the proposed method might be sensitive to the exploration probability $\epsilon$, given that it varies considerably across different environments. Is your algorithm robust to changes in $\epsilon$?

**Questions:**

- In Theorem C.10, how large is the condition number $\kappa$ typically, or in the worst case?


- Since the proposed method uses parallel models, it relates closely to ensemble-based algorithms, such as Bootstrapped DQN (Osband et al., 2016, [1]). Additionally, recent work on linear ensembles for bandits (Janz et al., 2024 [2]), published in NeurIPS 2024, is also relevant. Comparing your approach with these ensemble-based algorithms would enhance the paper.

    [1] Osband et al., Deep Exploration via Bootstrapped DQN, NeurIPS 2016.
    [2] Janz et al., Ensemble sampling for linear bandits: small ensembles suffice, NeurIPS 2024.

I will consider raising the score once my concerns and questions have been addressed.

---

### Official Review · Reviewer_oGqe · 2024-11-03

**Soundness:** 2
**Presentation:** 3
**Contribution:** 1
**Rating:** 3
**Confidence:** 5

**Summary:**

The paper proposes **Practical ε-Exploring Thompson Sampling (PETS)**, aiming to enhance exploration in reinforcement learning (RL) with high-dimensional continuous action spaces. The main contributions are:

- **Application of Thompson Sampling (TS)** to continuous control tasks by addressing computational intractability.
- **Use of Langevin Monte Carlo (LMC)** for approximate posterior sampling of the action-value function parameters.
- **Maintenance of ensembles of parallel Markov chains** to reduce sample correlation and improve exploration diversity, effectively approximating the posterior distribution.
- Introduction of both **gradient-based and gradient-free optimization methods** to approximate the optimal action under sampled models.
- **Regret analysis** showing that PETS achieves regret bounds comparable to the best-known results in discrete settings.

Empirical results indicate that integrating PETS with existing RL algorithms like POMP, MBPO, and SAC leads to improved performance and stability on benchmark continuous control tasks.


- **Lack of Novelty**: The method closely mirrors ensemble sampling techniques without proper acknowledgment or differentiation, undermining claims of innovation.

- **Omission of Related Work**: Fails to discuss relevant literature on ensemble methods and randomized value functions, which is critical for situating the contribution within the field.

**Strengths:**

- **Addresses a Relevant Challenge**: Tackles the problem of exploration in high-dimensional continuous action spaces, which is important in RL research.

- **Integration with Existing Algorithms**: Demonstrates that PETS can be incorporated into existing RL frameworks, potentially offering improvements without significant modifications.

**Use of LMC and Parallel Markov Chains**: Leveraging LMC for posterior sampling is appropriate due to the high dimensionality of the action space. Maintaining multiple parallel Markov chains is an effective strategy to reduce sample correlation.

- **Empirical Performance**: PETS shows improved performance over baseline algorithms (e.g., a 38% improvement over POMP on Walker2d), suggesting effectiveness in enhancing exploration.

- **Exploration Effectiveness**: Visualizations and ablation studies support the claim that PETS improves exploration diversity.

- **Theoretical Analysis**: Provides regret bounds that align with established results, supporting the method's theoretical validity.

**Weaknesses:**

- **Use of LMC and Parallel Markov Chains**:  This approach essentially amounts to maintaining an ensemble of models and uniformly sampling one at each decision point. This is conceptually similar to existing ensemble sampling methods used to approximate TS. [Line 7 of Algorithm 1]

- **Computational Complexity**: The paper lacks a thorough analysis of the per-episode computational complexity, especially concerning the number of episodes and the number of parallel Markov chains maintained. The overhead introduced by managing multiple chains and performing multiple optimizations per action selection could impact scalability in large-scale applications.

- **Connection to Ensemble Methods**: The methodology shows strong parallels to ensemble sampling approaches, such as those found in "Ensemble Langevin DQN" by Dwaracherla & Van Roy, where ensembles are used with Langevin dynamics for exploration. The paper does not acknowledge or discuss these similarities, missing an opportunity to position PETS within the broader context of ensemble-based RL methods.

---

- **Statistical Significance and Computational Cost**:

  - **Statistical Analysis**: The experiments are conducted over a limited number of seeds (5 to 8), which may not provide sufficient statistical power to confirm the results' significance.

  - **Computational Overhead**: The paper does not quantify the additional computational cost per episode introduced by maintaining multiple parallel Markov chains and performing optimization steps. This omission makes it difficult to assess the practical trade-offs between performance gains and computational efficiency.

- **Baseline Comparisons**: The experimental comparisons are primarily against base algorithms without PETS. The paper does not compare PETS against Ensemble Langevin DQN [https://arxiv.org/abs/2002.07282]

---

**Related Work Comparison**

The paper contributes to the application of TS in continuous action spaces but fails to adequately discuss relevant existing literature, particularly on ensemble methods:

- **Ensemble Methods and Approximate TS, Randomized Value Functions**:

  - **"Ensemble Sampling" by Lu & Van Roy**: This work develops ensemble sampling as an approximation to TS, maintaining an ensemble of models and sampling from them to drive exploration. PETS's use of multiple parallel Markov chains closely resembles this approach but lacks acknowledgment of this connection.

  - **"Ensemble Langevin DQN" by Dwaracherla & Van Roy**: This paper demonstrates that ensemble methods combined with Langevin dynamics can achieve deep exploration without additional complexity for epistemic uncertainty representation. PETS uses a similar strategy but does not reference this work.

  - **"Deep Exploration via Bootstrapped DQN" by Osband et al.**: Introduces an ensemble-based method (Bootstrapped DQN) for deep exploration in RL, which PETS could be compared against.

  - **"Deep Exploration via Randomized Value Functions" by Osband et al.**: Discusses using randomized value functions to encourage exploration, which is conceptually related to PETS's approach.

  - **"Randomized Value Functions via Multiplicative Normalizing Flows" by Touati et al.**: Proposes achieving randomized value functions in high-dimensional domains using variational Bayesian methods, offering an alternative to PETS's LMC sampling.

  - **"Q-Star Meets Scalable Posterior Sampling" by Li et al.**: Presents an efficient method for approximate posterior sampling in RL, achieving sublinear regret and superior performance in large-scale deep RL benchmarks, which could provide valuable insights or comparisons for PETS.

The omission of these works limits the paper's positioning within the existing body of research and undermines its claim of novelty.

---

**5. Clarity and Presentation**

The paper is structured logically, but several aspects could be improved:

- **Methodological Clarity**:

  - **Details on Computational Complexity**: A detailed analysis of the per-episode computational complexity is missing. Understanding how the number of episodes and the number of parallel Markov chains impact computational resources is crucial for practical applications.

  - **Algorithmic Similarities**: The paper does not clarify how its approach differs from or improves upon existing ensemble methods, which could confuse readers familiar with ensemble sampling.

- **Algorithm Descriptions**:

  - **Algorithm 1**: Line 7's operation of sampling from multiple models is similar to ensemble sampling, but this is not acknowledged, leading to potential misunderstandings about the novelty of the approach.

- **Presentation of Theoretical Results**:

  - **Regret Analysis**: The theoretical analysis may be dense and could benefit from additional explanations or simplifications to enhance accessibility.

---

**6. Overall Impact**

While PETS aims to advance exploration strategies in RL with continuous controls, its impact is limited by several factors:

- **Novelty**: The core methodology closely resembles ensemble sampling approaches already present in the literature. Without acknowledging or differentiating from these methods, the paper's originality is diminished.

- **Practical Contribution**: The lack of computational complexity analysis raises concerns about the method's scalability and practical applicability in large-scale RL tasks.

- **Theoretical and Empirical Validation**: Although the paper provides theoretical regret bounds and empirical improvements, these are less compelling without thorough comparisons to existing ensemble-based exploration methods.

---

**Weaknesses:**

- **Lack of Novelty**: The method closely mirrors ensemble sampling techniques without proper acknowledgment or differentiation, undermining claims of innovation.

- **Omission of Related Work**: Fails to discuss relevant literature on ensemble methods and randomized value functions, which is critical for situating the contribution within the field.

- **Insufficient Computational Analysis**: Does not provide an analysis of the computational complexity per episode, leaving practical efficiency and scalability unaddressed.

- **Limited Experimental Rigor**: Experiments lack statistical significance testing and do not compare PETS against other state-of-the-art exploration methods, weakening the empirical validation.

**Suggestions for Improvement:**

1. **Acknowledge and Discuss Related Work**: Incorporate discussions of ensemble sampling methods, such as Bootstrapped DQN, Ensemble Sampling, and Ensemble Langevin DQN. Highlight how PETS relates to, differs from, and potentially improves upon these approaches.

2. **Clarify Novel Contributions**: Clearly articulate what is novel about PETS in comparison to existing ensemble methods. If PETS offers specific advantages or innovations, these should be explicitly stated and justified.

3. **Analyze Computational Complexity**: Provide a detailed analysis of the per-episode computational cost, including how it scales with the number of episodes and parallel Markov chains. Discuss any trade-offs between computational overhead and performance gains.

4. **Enhance Experimental Evaluation**:

   - **Statistical Significance**: Increase the number of random seeds and perform appropriate statistical tests to strengthen the reliability of the results.

   - **Comparative Baselines**: Include comparisons with other ensemble-based exploration methods and state-of-the-art algorithms to contextualize the performance improvements.

5. **Improve Clarity and Presentation**:

   - **Methodological Details**: Expand on the implementation aspects, particularly regarding the optimization methods and how parallel chains are maintained.

   - **Algorithm Descriptions**: Update algorithm pseudocode to reflect similarities and differences with existing methods, and clearly explain any unique steps.

6. **Theoretical Insights**: Simplify the theoretical analysis where possible, and provide intuitive explanations to make the content accessible to a broader audience.

By addressing these issues, the paper could significantly strengthen its contribution to the field, offering clearer insights into how PETS advances exploration in reinforcement learning and distinguishing it from existing ensemble methods.

**Questions:**

1. **Relationship to Ensemble Methods:**

   - **Question:** How does your method differ from existing ensemble-based approaches, such as Bootstrapped DQN, Ensemble Sampling, and Ensemble Langevin DQN?
   - **Suggestion:** Please clarify the distinctions between PETS and these ensemble methods. Specifically, explain how maintaining multiple parallel Markov chains in PETS offers advantages over traditional ensembles used in approximate Thompson Sampling.

2. **Acknowledgment of Related Work:**

   - **Question:** Why are key related works on ensemble sampling and randomized value functions omitted from the paper?
   - **Suggestion:** Including discussions of relevant literature, such as:
     - "Deep Exploration via Bootstrapped DQN" (Osband et al.)
     - "Ensemble Sampling" (Lu & Van Roy)
     - "Ensemble Langevin DQN" (Dwaracherla & Van Roy)
     - "Deep Exploration via Randomized Value Functions" (Osband et al.)
     - "Q-Star Meets Scalable Posterior Sampling" (Li et al.)
     This will position your work within the broader context and highlight its contributions compared to existing methods.

3. **Novel Contributions:**

   - **Question:** Can you explicitly state the novel contributions of PETS compared to existing ensemble sampling and randomized value function methods (including the LMC based randomized value)?
   - **Suggestion:** Highlight any unique aspects of PETS, such as theoretical improvements, practical benefits, or empirical performance gains that set it apart from similar approaches.

4. **Per-Episode Computational Complexity:**

   - **Question:** How does the computational complexity of PETS scale with the number of episodes and the number of parallel Markov chains (`n_samples`)?
   - **Suggestion:** Provide a detailed analysis of the per-episode computational complexity. Discuss the trade-offs between the computational overhead of maintaining multiple chains versus the performance improvements achieved.

5. **Algorithmic Details and Clarity:**

   - **Question:** In Algorithm 1, line 7 involves uniformly selecting a model from multiple posterior samples, which resembles ensemble sampling approximations to Thompson Sampling. How does this differ from existing ensemble methods?
   - **Suggestion:** Elaborate on this step, clarifying how your approach is distinct. If PETS introduces specific innovations in how the ensembles are generated or used, make this explicit in the methodology section.

6. **Use of Langevin Dynamics in Ensembles:**

   - **Question:** Since ensemble methods have been used with Langevin-type randomized value functions (e.g., "Ensemble Langevin DQN"), how does PETS improve upon or differ from these techniques?
   - **Suggestion:** Compare PETS to methods like "Ensemble Langevin DQN" by Dwaracherla & Van Roy. Discuss whether PETS offers theoretical or practical advantages, such as better exploration efficiency or ease of implementation.

7. **Experimental Comparisons:**

   - **Question:** Have you compared PETS to other state-of-the-art exploration methods, particularly ensemble-based approaches?
   - **Suggestion:** Include experimental results comparing PETS with algorithms like Bootstrapped DQN and Ensemble Sampling. This would strengthen the empirical evaluation and demonstrate PETS's effectiveness relative to existing methods.

8. **Statistical Significance of Results:**

   - **Question:** Are the performance improvements observed in your experiments statistically significant?
   - **Suggestion:** Increase the number of random seeds used in experiments and perform statistical significance testing. This will bolster the credibility of your results and address concerns about variability due to random initialization.

9. **Clarity on Computational Overhead:**

   - **Question:** What is the practical impact of maintaining multiple parallel Markov chains on computation time and resources?
   - **Suggestion:** Provide empirical measurements of the computational overhead introduced by PETS. Discuss how this overhead scales with `n_samples` and the implications for real-world applications.

10. **Theoretical Analysis Accessibility:**

    - **Question:** Can you provide more intuitive explanations of the theoretical regret analysis to make it accessible to a wider audience?
    - **Suggestion:** Include examples or diagrams that illustrate key theoretical concepts. Simplify complex mathematical derivations where possible, or provide additional explanatory text in appendices.

11. **Future Work and Limitations:**

    - **Suggestion:** Acknowledge any limitations of PETS, such as scalability issues or environments where it may perform less effectively. Outline potential future research directions to address these challenges.

12. **Implementation Details:**

    - **Question:** Could you provide more detailed information on implementing PETS, especially regarding the maintenance of parallel chains and integration with existing algorithms?
    - **Suggestion:** Including pseudo-code snippets, parameter settings, or practical tips would help practitioners replicate and build upon your work.

---

### Note · Authors · 2024-11-21

I have read and agree with the venue's withdrawal policy on behalf of myself and my co-authors.